# Utilizing Imine Bonds to Create a Self-Gated Mesoporous Silica Material with Controlled Release and Antimicrobial Properties

**DOI:** 10.3390/nano13081384

**Published:** 2023-04-16

**Authors:** Yuyang Lu, Xutao Li, Jiaqi Xu, Huimin Sun, Jie Sheng, Yishan Song, Yang Chen

**Affiliations:** 1College of Food Science and Technology, Shanghai Ocean University, Shanghai 201306, China; 2NEST Laboratory, Department of Physics, Department of Chemistry, College of Sciences, Shanghai University, Shanghai 200444, China; 3Shanghai Yaolu Instrument & Equipment Co., Ltd., Shanghai 200444, China

**Keywords:** self-gating, silica nanomaterials, pH-sensitive, imine bond, antimicrobial

## Abstract

In recent years, silica nanomaterials have been widely studied as carriers in the field of antibacterial activity in food. Therefore, it is a promising but challenging proposition to construct responsive antibacterial materials with food safety and controllable release capabilities using silica nanomaterials. In this paper, a pH-responsive self-gated antibacterial material is reported, which uses mesoporous silica nanomaterials as a carrier and achieves self-gating of the antibacterial agent through pH-sensitive imine bonds. This is the first study in the field of food antibacterial materials to achieve self-gating through the chemical bond of the antibacterial material itself. The prepared antibacterial material can effectively sense changes in pH values caused by the growth of foodborne pathogens and choose whether to release antibacterial substances and at what rate. The development of this antibacterial material does not introduce other components, ensuring food safety. In addition, carrying mesoporous silica nanomaterials can also effectively enhance the inhibitory ability of the active substance.

## 1. Introduction

Foodborne pathogens not only cause food spoilage but also pose a serious threat to human health, making them the “number one enemy” of food safety [1]. In China and across the world, foodborne diseases caused by these pathogens are among the top food safety concerns. Common foodborne pathogenic bacteria such as *E. coli*, *S. aureus*, *Salmonella*, *Listeria*, and *Vibrio parahaemolyticus* are often found in meat, aquatic products, soybean products, vegetables, dairy products, and fruit juices. Consumption of food contaminated with these pathogens can lead to discomfort and even death in severe cases [2,3].

Food safety issues caused by foodborne pathogens have been a continuous concern, highlighting the need for the development of antimicrobial agents [4]. Such agents are vital in preventing foodborne diseases by inactivating or inhibiting pathogenic microorganisms [5]. Antibacterial agents can be divided into three categories: inorganic, organic, and natural [6]. Inorganic and organic antibacterial agents have good antimicrobial properties but come with significant drawbacks such as high cost, poor heat resistance, easy hydrolysis, and certain toxicity. However, natural antibacterial agents are derived mainly from natural substances in plants [7]. They are suitable for use in the food industry because of their high antibacterial efficiency, safety, and nontoxicity [8]. However, natural antibacterial agents have clear disadvantages such as easy volatility, poor thermal stability, and strong, irritating odors, which limit their use in the food industry [9]. Hence, it is necessary to improve the physical properties of natural antibacterial agents while maintaining or enhancing their antibacterial activity by using technical means such as loading them into stable inorganic matrices [10].

Mesoporous silica nanomaterials (MSN) have emerged as a highly efficient and promising carrier for natural antimicrobial agents due to their adjustable pore size, large specific surface area, ease of surface modification, high thermal stability, controllable structure, and excellent biocompatibility [11,12]. Carbohydrates, organic molecules, peptides, and proteins have been employed as responsive “gating factors” or capping agents in current MSN systems [13,14]. However, despite their significant advantages, most of these systems still face challenges such as a lack of studies on the toxicity of capping agents and limited bioresponsiveness, especially when used for bacteriostasis in food [15].

In view of the above research status, we constructed a simple antimicrobial “citral-self-gated” MSN antimicrobial nanosystem by introducing pH-sensitive imine covalent bonds, that is, using the chemical properties of citral to make the antibacterial system achieve a self-controlled release rate. The main aim of this study was to employ the antimicrobial agents themselves as pH-sensitive “gating factors” to minimize the possible risks of auxiliary capping agents, and the active part of their molecular structure can better prevent the growth of foodborne pathogenic bacteria [16]. Our design involved using citral (CIT) as both the antibacterial agent and “gating factor” due to its excellent antibacterial performance and aldehyde group [17,18]. A hollow mesoporous silica sphere functionalized by an amino group was loaded with CIT and then gated by the CIT on pore outlets via the formation of the imine covalent bond between the CIT and amino group. In the event of food being infected with pathogenic bacteria (pH ≤ 6.8), the loaded CIT would be released on demand due to the hydrolysis of imine bonds induced by a pH drop [15]. When the food system is not infected, the antimicrobial vector will remain stable [19,20]. The synergistic action of the CIT, amino groups, and mesoporous silica nanomaterial resulted in a pH-sensitive, controllable, and long-acting antibacterial performance, making this carrier highly suitable for antibacterial applications and the design of new “self-gated” antibacterial systems.

In conclusion, our study demonstrates the potential of MSN as a highly efficient carrier for natural antimicrobial agents and provides a new strategy for the development of self-gated antibacterial systems. The results of this study can be used as a reference in the field of antibacterial applications, especially for food antimicrobial agents (Figure 1) [13,21].

## 2. Materials and Methods

### 2.1. Materials

Materials including 2,6-dimethyl-2,6-octadiene aldehyde (citral, 97%), tetraethyl orthosilicate (TEOS, 98%), cetyltrimethylammonium bromide (CTAB, 99%), aqueous ammonia (28%), 3-aminopropyltriethoxysilane (APTES, 97%), absolute ethanol (99.8%), hydrochloric acid (10%), 2,4-dinitrophenylhydrazine test solution (98%), trypsin soybean broth (TSB), and phosphate buffer saline (PBS, 0.1 M, pH = 7.4) were all provided by Sangon Biotech (Shanghai, China).

### 2.2. Synthesis of Hollow Mesoporous Silica Spheres (HMSS)

In this experiment, CTAB (0.9 g) was dissolved in a mixture of distilled water (300 mL) and ethanol (180 mL). To this solution, ammonia (8 mL) was added while stirring continuously for one hour. Then, TEOS (6 g) was slowly added to the solution drop by drop and stirred at 35 °C for 6 h. The resulting solid was recovered, washed with deionized water, and dried overnight at 60 °C.

Next, the solid (0.5 g) was added to deionized water (250 mL) and heated to 90 °C for 24 h to synthesize the hollow structure. The solid was collected by filtration, washed with deionized water, and dried under high vacuum overnight. Finally, the sample was refluxed with ethanol in a Soxhlet extractor for 2 days to remove the template, resulting in the formation of the HMSS [22,23].

### 2.3. Fabrication of Amino-Modified HMSS (M-NH_2_)

A white suspension was obtained after HMSS (1 g), added to 40 mL of methanol, and stirred until uniformly dispersed. Thereafter, APTES (0.8 mL) was dropped into the white suspension and refluxed at 120 °C for 16 h under nitrogen atmosphere. Finally, the product (M-NH_2_) was washed with ether and dichloromethane as a co-solvent and dried overnight in high vacuum at 60 °C.

### 2.4. Loading and Capping of Citral (CIT) on M-NH_2_ (M-NH_2_-CIT@CIT) and HMSS (M-CIT)

By soaking M-NH_2_ (1 g) in 40 mL anhydrous ethanol containing 1 mL CIT and stirring at pH 6.0 and room temperature for 24 h, an imine bond is formed between the aldehyde group of CIT and the amino group of M-NH_2_ at pH 8.0. After centrifugation and washing, the final product is designated as M-NH_2_-CIT@CIT [18].

HMSS (1 g) was immersed in an ethanol solution (40 mL) containing CIT (1 mL) and stirred at room temperature for 24 h to synthesize M-CIT. Finally, the product was centrifuged, washed, and dried.

### 2.5. Characterizations

The HMSS, M-NH_2_, and M-NH_2_-CIT@CIT were analyzed using standard techniques, including scanning electron microscopy (SEM), Fourier-transform infrared spectroscopy (FT-IR), thermogravimetric analysis (TGA), N_2_ adsorption/desorption isotherms, and a Zeta-potential analyzer.

The silica morphology was obtained by SEM (Hitachi SU5000, Tokyo, Japan) at 6 kV acceleration voltages, while the chemical composition of silica carriers was analyzed by FT-IR r (Nicolet Instrument, Thermo Fisher, Waltham, MA, USA). Thermal stability analysis of silica particles was conducted using a TGA (STA449C/4/G, Netzsch, Selb, Germany) at a heating rate of 10 K min^−1^ in an air-filled condition (80 mL min^−1^) at a temperature of 30 to 800 °C. N_2_ adsorption and desorption isotherms were obtained using an automatic adsorption analyzer (BELSORP-mini II, Bengaluru, India) to analyze the specific surface area, pore diameter and volume. The Zeta potential was measured using a Zeta-potential analyzer (Zetasizer Nano ZS90, Malvern Instruments, Malvern, UK) at 25 °C. 

### 2.6. Plotting CIT Standard Curve

Wavelength determination: According to the literature, the color reaction of 2,4-dinitrophenylhydrazine with aldehydes and ketones is commonly used for the qualitative analysis of these compounds [22]. First, the CIT standard solution (1 mL) with a concentration of 0.4 mg/mL was accurately absorbed into a test tube. The 2,4-dinitrophenylhydrazine (5 mL) test reagent was added simultaneously, and the reaction was immediately mixed for 20 min. The absorbance at the wavelength of 450–500 nm was then determined by spectrophotometry. The results showed that the reaction solution had better absorbance at a wavelength of 460 nm.

To plot the standard curve of the reaction solution, CIT (50 mg) was placed in a 50 mL brown volumetric flask and then shaken with anhydrous ethanol. The CIT concentration was diluted to 0.5, 0.4, 0.3, 0.2, and 0.1 mg/mL in test tubes with ethanol. Each of the above concentrations of solution (1 mL) was taken in a test tube, and 2,4-dinitrophenylhydrazine (5 mL) was added, followed by a 20 min reaction at room temperature. Finally, the absorbance of the above solutions was measured by spectrophotometry at a wavelength of 460 nm to plot the standard curve of the reaction solution.

### 2.7. pH-Dependent CIT Release

To investigate the pH response mechanism of foodborne microbial contamination, we selected *E. coli* as a representative and inoculated it in TSB to observe changes in the environmental pH value. In addition, the loading of CIT has been determined by thermogravimetric analysis (TGA) curve, and the pH-responsive release characteristics of M-NH_2_-CIT@CIT and M-CIT have been studied.

Specifically, we dispersed 5 mg of CIT-containing M-NH_2_-CIT@CIT and M-CIT in a 10 mL ethanol solution with a pH of 7.1, and adjusted the pH of the solution to 6.8 and 5.8 by adding diluted HCl dropwise. We then transferred the centrifuge tube to an orbital water bath and shook it at 120 rpm at 37 °C and collected appropriate amounts of solution every two hours. Finally, the absorbance of these solutions was measured at 460 nm by using a spectrophotometer, and we calculated the release concentration of CIT in these materials by fitting the data to the standard curve [24].

### 2.8. Culture Conditions and Bacterial Strain

*E. coli* (Gram-negative) was obtained from Shanghai Ocean University. All strains were activated and stored at −80 °C, then transferred to tryptic soy agar (TSA) supplemented with soybean and kept in a 4 °C refrigerator for later use. When needed, colony cells were extracted from the solid medium and transferred to 10 mL of tryptic soy broth (TSB). After 24 h of cultivation at 37 °C, the inoculum of approximately 10^8^ colony-forming units (CFU) per mL was obtained for subsequent experiments.

### 2.9. Antibacterial Activity Assays In Vitro

The antimicrobial effects of M-CIT, M-NH_2_-CIT@CIT, CIT (the concentration of CIT was 1 mg/mL), and control groups (HMSS and M-NH_2_) were determined using the optical density (OD) growth curve of *E. coli*. After inoculating TSB with 0.1 mL bacterial solution, the prepared antimicrobial material was added and cultured at 37 °C for 24 h. Finally, the absorbance of treated *E. coli* was measured with a microplate reader at 600 nm [25]. Each condition was prepared in triplicate, and all experiments were repeated three times.

The minimum inhibitory concentration (MIC) of M-CIT and M-NH_2_-CIT@CIT was determined under conditions of 4, 2, 1, 0.5, and 0.25 mg/mL, which is the concentration of the biologically active compound that causes a 90% inhibition of *E. coli* within 24 h. Based on the previous thermogravimetric (TG) results, the amount of M-CIT and M-NH_2_-CIT@CIT required to provide these concentrations was calculated. *E. coli* microbial density was diluted to 1 × 10^6^ cells/mL using TSB. Next, different amounts of free CIT and M-NH_2_-CIT@CIT were added to a conical flask containing 15 mL of TSB and mixed. Then, a bacterial suspension of 10 µL was inoculated, and the final bacterial concentration in the conical flask was 1 × 10^4^ cells/mL. The mixture was cultured at 37 °C and 180 rpm for 24 h. After incubation, the number of viable cells was measured using the plate counting method as colony-forming units (CFUs) [26,27]. These values were logarithmically transformed and expressed as log CFU/mL. All treatments were repeated three times. Positive (add M-CIT and M-NH_2_-CIT@CIT) and negative (nothing is added) controls were included in all tests. The antibacterial rate (X) was calculated using the equation as follows:$$X=\left(1-\frac{B}{A}\right)\times 100\%$$

*A*: Number of colonies in the control group*B*: Number of sample colonies.

### 2.10. Sterilization Mechanism

Scanning electron microscopy was used to observe the morphological changes of *E. coli* after treatment with M-NH_2_-CIT@CIT [28]. The process involved adding the carrier to the *E. coli* suspension, incubating it for 24 h at 37 °C, and then centrifuging the sample. The supernatant was discarded, and the bacterial solution was fixed using glutaraldehyde. The sample was then washed with a PSB buffer and dehydrated using ethanol solutions of different concentration gradients. Dehydrated cells were freeze-dried for 24 h and observed under SEM to detect any morphological changes.

### 2.11. Data Processing

All experiments were performed in triplicate. Origin 9.0 software was used to make diagrams. The data are presented as the mean ± SD (n = 3). Differences between means were tested by test. Differences were defined as significant at *p* ≤ 0.05.

## 3. Results and Discussion

### 3.1. Characterization Results

Zeta potential is a critical parameter for predicting the surface charge of nanomaterials [29]. The surface charge has a significant impact on the physical stability of materials in suspension, as well as the preparation and interaction of materials with bacteria. As shown in Table 1, HMSS has a negative charge due to the abundance of silica hydroxyl groups on its surface. However, after modification with positively charged APTES, the Zeta potential of M-NH_2_ and M-NH_2_-CIT@CIT became positive. This renders the prepared materials capable of electrostatically adsorbing to most bacteria. The Zeta potential of M-NH_2_ shifts from −12 mV to 25.4 mV (Figure 1), while that of M-NH_2_-CIT@CIT was 24.5 mV. This indicates that the Zeta potential of M-NH_2_ and M-NH_2_-CIT@CIT is close to the minimum Zeta potential (±30 mV) required for the physical stability of nanoparticles in suspension [30]. Therefore, the functionalized materials are expected to be physically stable due to electrostatic repulsion and are unlikely to agglomerate.

The SEM image in Figure 2 reveals the difference in size and morphology between HMSS and M-NH_2_-CIT@CIT. As shown in Table 1, the average particle size of HMSS was 616 nm, while the average particle sizes of functionalized M-CIT and M-NH_2_-CIT@CIT were 627 and 634 nm, respectively. All materials exhibit a desirable morphology and uniform particle size distribution, and their unique spherical hollow structure offers a high specific surface area while ensuring high CIT load. Some fine particles can be seen on the surface of HMSS after functionalization by CIT, which slightly increases its surface roughness. This discovery demonstrates that the synthesis of M-NH_2_-CIT@CIT does not affect the overall morphology and mesoporous structure of HMSS, indicating the high stability of hollow mesoporous silica spheres [31,32]. This finding is consistent with the literature [33].

The infrared spectrum of HMSS in Figure 3a shows a strong and wide absorption band at 1084 cm^−1^, which corresponds to the anti-symmetric stretching vibration of Si–O-Si. The band at 790 cm^−1^ corresponds to the symmetric stretching vibration of the Si-O bond, and the band at 958 cm^−1^ belongs to the bending vibration absorption band of Si-OH [34,35]. For the spectrum of M-NH_2_, except for the characteristic bands of HMSS, a new broad band appears at 3000 to 3600 cm^−1^ corresponding to the loading of N-H [36]. However, the simultaneous presence of the antisymmetric vibrational band of O-H within 3000–3600 cm^−1^ may mask the band of N-H. At the same time, the presence of characteristic bands corresponding to methylene C-H was observed at 2925 and 2854 cm^−1^ in the spectra of M-NH_2_ and M-NH_2_-CIT@CIT, which could equally indicate the successful modification of APTES on HMSS [37]. In the spectrum of CIT, the absorption bands at 1721 and 1683 cm^−1^ are attributed to the stretching vibration of C = O, and the absorption bands at 1418 and 1233 cm^−1^ are attributed to the coupling of OH in-plane bending vibration and C-O stretching vibration [38]. The absorption bands at 936 cm^−1^ are attributed to the -OH out-of-plane bending vibration. For the M-CIT spectrum, the characteristic band corresponding to CIT was found at 1418 and 1233 cm^−1^, and the characteristic band corresponding to HMSS was also found at 1084 and 958 cm^−1^, indicating the successful preparation of M-CIT. For M-NH_2_-CIT@CIT spectra, in addition to the characteristic band of M-NH_2_, the characteristic band of CIT is also observed. These observations demonstrate the successful synthesis of M-NH_2_-CIT@CIT.

Thermogravimetric analysis (TGA) diagrams of HMSS, M-CIT, M-NH_2_, and M-NH_2_-CIT@CIT at various stages are shown in Figure 3b. Due to the removal of water on the HMSS surface, the curve only shows a peak corresponding to weight loss in the temperature range of 30–120 °C [39]. Its mass loss was 8.69%. Additionally, two peaks have been observed in the thermogravimetric curve of M-NH_2_. The first peak was in the temperature range of 30–120 °C, which was consistent with HMSS. The second peak emerged in the temperature range of 120–400 °C, which was due to the decomposition of APTES [40]. This phenomenon proved that HMSS was successfully amino functionalized. M-CIT also has two peaks. However, the tendency of its weight loss in the temperature range of 30–120 °C is very large, which may be due not only to the evaporation of water, but also to the thermal degradation of encapsulated CIT. When the temperature exceeds 120 °C, the residual CIT is continuously thermally degraded, but the trend of weight loss is not obvious. Finally, the TG curve of M-NH_2_-CIT@CIT presents two characteristic peaks of weight loss; the first is in the temperature range of 30–120 °C, which is also caused by water evaporation and thermal degradation of CIT. The second peak occurs in the temperature range of 120–800 °C, which is the result of thermal degradation of CIT and APTES [41]. The above findings proved the successful preparation of M-CIT, M-NH_2_, and M-NH_2_-CIT@CIT. Meanwhile, comparing these results, the CIT load in M-CIT and M-NH_2_-CIT@CIT was 19% and 32%, respectively. M-CIT has a much smaller CIT-load than M-NH_2_-CIT@CIT, probably caused by the inability of M-CIT to gate CIT. The results showed that the preparation method significantly increased the loading capacity.

The nitrogen adsorption/analysis isotherms of HMSS, M-NH_2_, and M-NH_2_-CIT@CIT all exhibit a type-IV curve, indicating the presence of mesoporous structures [35,42]. The nitrogen adsorption isotherms of HMSS show a significant increase in the range of P/P0 = 0.2–0.4 due to capillary condensation in the pores of the material [43]. However, the isothermal adsorption lines of M-NH_2_ and M-NH_2_-CIT@CIT do not exhibit a similar increase, indicating that amino functionalization, and CIT loading reduced capillary condensation during nitrogen adsorption in the pores [44,45]. Compared with HMSS, both M-NH_2_ and M-NH_2_-CIT@CIT show a reduction in specific surface area, pore diameter, and pore volume, as shown in Figure 4b and Table 1. The specific surface areas of HMSS, M-NH_2_, and M-NH_2_-CIT@CIT are 885.12 m^2^/g, 368.26 m^2^/g, and 268.73 m^2^/g, respectively. This reduction in specific surface area is due to the surface modification that caused blockage of material pores and verifies the successful introduction of APTES and CIT. The significant difference in specific surface area between HMSS and M-NH_2_-CIT@CIT indicates that the antibacterial material has excellent CIT-loading capacity. Fan Gao et al. also found similar results in their studies on the preparation of amino-functionalized mesoporous silica [27].

### 3.2. The Determination of CIT Standard Curve

Figure 5a illustrates the color reaction of CIT with 2,4-dinitrophenylhydrazine. As shown in the figure, 2,4-dinitrophenylhydrazine is an effective color development agent for CIT. Figure 5b demonstrates that the reaction solution exhibits improved absorbance when measured at a wavelength of 460 ± 1 nm.

The standard curve for the CIT solution is shown in Figure 5c. According to the fitting, the calculated formula of the standard curve was y = −0.0129 + 1.549x, R^2^ = 0.9983 [46].

### 3.3. pH-Dependent Drug Release

To investigate the gated effect of M-NH_2_-CIT@CIT, in vitro drug release experiments were performed on M-NH_2_-CIT@CIT and the control group M-CIT at different pH values. M-CIT lacks the amino group necessary to seal CIT, and as expected, released more than 60% of CIT within 12 h at pH 7.1 (Figure 6b). In contrast, M-NH_2_-CIT@CIT at pH 7.1 released less than 10% of CIT within 24 h, demonstrating excellent gating effect under physiological conditions (Figure 6a). Furthermore, the release amount and rate of CIT from M-NH_2_-CIT@CIT varied significantly with pH changes. At pH 6.8, over 50% of CIT was released within 12 h, followed by a slower release rate. Upon further acidification to pH 5.8, a new round of explosive CIT release occurred, reaching over 80% release within 12 h due to the breaking of imine bonds [47,48,49]. These results confirm the highly efficient gating effect of imine bonds and demonstrate the multifunctional strategy of the “CIT-self-gated” approach [50,51], which allows on-demand release at different pH levels without the use of auxiliary gating factors [52,53].

Figure 7 illustrates the pH changes of *E. coli* growing in TSB. The pH of *E. coli* dropped from 7.1 to 6.8 within two hours, indicating that the antibacterial material M-NH_2_-CIT@CIT rapidly released CIT and exerted antibacterial effects within the first two hours of growth and reproduction of *E. coli* (as concluded in Section 3.4). When *E. coli* reached 10 h of growth, the pH of its environment began to rise, possibly due to metabolic substances produced in the late growth stage of the bacteria.

### 3.4. Analysis of Antibacterial Activity In Vitro

Figure 8 depicts the growth curve of *E. coli* after exposure to various materials. As is commonly known, longer growth cycles and lower OD values indicate superior antibacterial activity. Upon comparing the OD growth curve of *E. coli*, the curve for HMSS showed a slight decline, which could be attributed to the agglomeration of the HMSS suspension. Similarly, the curve for M-NH_2_ also showed a marginal drop. This may be attributed not only to the positive surface charge of M-NH_2_, which attracts bacteria, but also to its relatively stable suspension [54]. Meanwhile, the curves for M-CIT and M-NH_2_-CIT@CIT were significantly lower than those for *E. coli* and even lower than that for CIT. This demonstrates the exceptional antimicrobial activity of M-CIT and M-NH_2_-CIT@CIT, which also enhances the antimicrobial activity of CIT. After 16 h, the curves for CIT and M-CIT showed a noticeable upward trend. This may be due to the fact that CIT is not secured by “gating factors” and gradually decreases over time [55,56]. In contrast, the curve for the “CIT-self-gated” antibacterial material M-NH_2_-CIT@CIT did not change considerably within 24 h, validating its on-demand release capacity and high-efficiency antimicrobial properties.

In summary, M-NH_2_-CIT@CIT exhibits excellent antibacterial activity and long-term antimicrobial capacity due to its on-demand release feature. Hence, this antibacterial material not only possesses good antibacterial ability but also exhibits long-term antibacterial ability, making it a promising candidate for future research.

Comparative experiments were carried out on agar plates containing different concentrations of M-CIT and M-NH_2_-CIT@CIT to determine the MIC values of the materials [57]. As shown in Figure 9 (a_1_, a_2_) and (b_1_, b_2_), M-CIT and M-NH_2_-CIT@CIT showed good antibacterial activity with 100% antimicrobial rate at high concentration (4 mg/mL and 2 mg/mL). However, the antimicrobial rate of M-CIT was reduced to 85% when the concentration was reduced to 1 mg/mL, indicating that the MIC value of M-CIT was about 1–2 mg/mL (Figure 10). Similarly, when the concentration of M-NH_2_-CIT@CIT is reduced to 1 mg/mL and 0.5 mg/mL, the antimicrobial rate is 95% and 90%, respectively, and the change is not obvious (Figure 10). Therefore, the MIC value of M-NH_2_-CIT@CIT is about 0.5–1 mg/mL, slightly lower than the MIC value of M-CIT. This indicates that the “CIT-self-gated” system of M-NH_2_-CIT@CIT not only has long-term antimicrobial performance, but also improves the antimicrobial efficiency of CIT.

In conclusion, the results of antimicrobial experiment demonstrate that the design of “CIT-self-gated” antimicrobial material M-NH_2_-CIT@CIT is successful.

### 3.5. Antibacterial Mechanism

SEM analysis was conducted to investigate the antibacterial mechanism of the “CIT-self-gated” antimicrobial material M-NH_2_-CIT@CIT on *E. coli*. The results, presented in Figure 11, show that the surface of *E. coli* cells was smooth and had complete cell membrane and cell wall before antibacterial treatment. However, after antimicrobial treatment with M-NH_2_-CIT@CIT, the surface of the cells became coarse and partially contracted. This phenomenon is consistent with the literature [27,58]. This effect can be attributed to the hydrophobic nature of CIT, which is an essential oil that can interact with lipids on bacterial cell membranes and increase the permeability of cell membranes. As a result, this causes cell leakage and affects bacterial activity [4,59].

## 4. Conclusions

This study demonstrated an ingenious “essential-oil self-gated” strategy. This pH-responsive bacteriostatic material is controlled by the bacteriostatic drug CIT itself, eliminating the need for complex “gating factors”. The characterization results demonstrate that this material not only enhances the stability and utilization rate of bacteriostatic drugs, but also attracts more bacteria due to its unique structure and surface charge. Compared to M-CIT, M-NH_2_-CIT@CIT achieves higher CIT load capacity as it is self-gated by CIT. This leads to reduced impact on food since less bacteriostatic material is needed for effective bacteriostasis. The “CIT-self-gated” system of M-NH_2_-CIT@CIT not only has long-term antimicrobial performance, but also improves the antimicrobial efficiency of CIT. This is due to the controlled release of CIT from the mesoporous matrix carrier in response to changes in pH and bacterial environment. The hydrophobicity of CIT allows it to interact with the bacterial cell membrane, leading to increased permeability and leakage of the cell contents. The SEM analysis further confirmed the damage to the bacterial cell membrane after treatment with M-NH_2_-CIT@CIT.

Overall, this new pH-responsive antimicrobial material has great potential in food preservation and other fields. The use of essential oil as a gating factor to regulate drug release is simple, efficient, and safe, making it an attractive alternative to traditional antimicrobial agents. Further research can be conducted to optimize the synthesis process and explore the application of this material in various fields.

## Data Availability

Data is contained within the article Materials.

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
