# Peer review of "Utilizing Imine Bonds to Create a Self-Gated Mesoporous Silica Material with Controlled Release and Antimicrobial Properties"

_nanomaterials, 2023, doi:10.3390/nano13081384_

Round 1
Reviewer 1 Report
The paper “Utilizing imine bonds to create a self-gated mesoporous silica material with controlled release and antimicrobial properties” by Lu et al. concerns the synthesis and characterization of new functionalized silica nanomaterials to be used as antibacterial agent for food pathogens. The paper is well written and organized. The experimental procedure is well defined and the research project is clear.
The main concern regards the absence of analysis or literature based discussion on mesoporous silica fate after food ingestion. The matrix degradation should be considered as well as potential toxicity after ingestion
Moreover, with a mean size of 600 nm is it still possible to considered them as nanomaterial?
Minor correction
Please change the term “weightlessness” with weight loss in TGA discussion
Please change “tg” with TG since tg is generally used to identify glass transition temperature
Please use the term band instead of peak in IR discussion
correct figure 7 with figure 10 in SEM description
I suggest to remove the term Moreover at the beginning of Conclusion section
Reviewer 2 Report
The manuscript by Sheng and coworkers describes the use of imine bonds in self-gated mesoporous silica for the controlled release and antimicrobial agents. The manuscript can be of interest to the journal after a major revision.
Some text is highlighted in yellow, please check.
Line 57-77: Examples of similar gates should be described in the introduction. This is important to put in context the developed work. The authors state that this is the first example of pH-responsive covalent bonding nanomaterial in the food antimicrobial, but no references to reviews or books are given. This needs to be carefully described. Also, examples of the chemistry reversibility of imine bonds with the pH should be clearly described in the introduction.
Line 85: Scheme 1: the chemical structures of the different elements should be described in the figure (imine bonds, etc.). The figure is confusing as it shows different types of nanoparticles. Does the system has one big nanoparticle with smaller nanoparticles inside?
Line 94 (material section): the text in this section should be revised and written according to the standard rules (i.e. indicating the quantities in brackets, etc.). In general, the destription of all methods should be improved and they should be described in more detail.
Lines 125-138: The x-ray diffraction powder pattern of the HMSS should be measured and reported.
Line 227, Table 1: check the red text, it should be black. In this table the quantity of CIT loaded in the nonoparticles should be included.
Line 244: Figure 2. It is not possible to see any porous in the hollow silica nanoparticles as represented schematically in Scheme 1. Please check.
Line 245: add "IR spectrum"
Line 342: Release studies. The release studies should be performed in the same timeframe to compare, i.e. 3 samples for 72 h at the 3 different pH values. It is not clear what is figure b (what is M-CIT)?
Line 244: Figure 7. An experiment using a buffer to control the pH of the bacteria should be performed and this should be used as a control to show that in these conditions of constant pH the M-NH2-CIT@CIT is not toxic.
Line 365: the legend of this figure is difficult to understand. Are all the control experiments performed at the same concentration? what is the difference between M-CIT and CIT?
Line 384: it is not clear what means a1, b1, etc. this needs to be clarified. In figure b, what are a and b?
Line 396. The results observed here need to be compared with similar results from the bibliography. Also, are the same results observed if the bacteria are treated with free CIT?
Line 399: the conclusions need to be improved by summarizing all the results from the paper (the present version only summarized some of the results)
Round 2
Reviewer 2 Report
The authors have replied and answered all the required changes, therefore I suggest the acceptance of the manuscript.